   

# Diallel panel reveals a significant impact of low-frequency genetic variants on gene expression variation in yeast

Andreas Tsouris[1,3], Gauthier Brach[1,3], Anne Friedrich [1], Jing Hou [1✉] & Joseph Schacherer [1,2✉]

## Abstract

**Unraveling the genetic sources of gene expression variation is essential to better understand the origins of phenotypic diversity in natural populations. Genome-wide association studies identified thousands of variants involved in gene expression variation, however, variants detected only explain part of the heritability. In fact, variants such as low-frequency and structural variants (SVs) are poorly captured in association studies. To assess the impact of these variants on gene expression variation, we explored a half-diallel panel composed of 323 hybrids originated from pairwise crosses of 26 natural *Saccharomyces cerevisiae* isolates. Using short- and long-read sequencing strategies, we established an exhaustive catalog of single nucleotide polymorphisms (SNPs) and SVs for this panel. Combining this dataset with the transcriptomes of all hybrids, we comprehensively mapped SNPs and SVs associated with gene expression variation. While SVs impact gene expression variation, SNPs exhibit a higher effect size with an overrepresentation of low-frequency variants compared to common ones. These results reinforce the importance of dissecting the heritability of complex traits with a comprehensive catalog of genetic variants at the population level.**

**Keywords** eQTL; Diallel; Low-frequency Variants; Structural Variants; Yeast
**Subject Categories** Chromatin, Transcription & Genomics; Genetics, Gene Therapy & Genetic Disease

## Introduction

Gene expression variation among individuals corresponds to an essential step linking genetic variation and phenotypic diversity observed in natural populations (Hill et al, 2021; Albertand Kruglyak, 2015, 15; Rockman and Kruglyak, 2006). Dissecting the genetic basis of gene expression variation at the population level is therefore crucial to better understand the genotype-phenotype relationship. Genetic variants or loci associated with gene expression variation (i.e., expression Quantitative Trait Loci, eQTL) have been detected using different mapping strategies, such as linkage and genome-wide association studies (GWAS) (Mackay et al, 2009). Large-scale transcriptomic surveys in model and non-model systems have highlighted the preponderance of genetic variants impacting gene expression variation between individuals (Zhang et al, 2022; Albert et al, 2018; Kita et al, 2017; Schadt et al, 2003; Battle et al, 2014; West et al, 2007; Zhang et al, 2011; Ferraro et al, 2020; GTEx Consortium, 2017; Kawakatsu et al, 2016; Vu et al, 2015; Rockman et al, 2010; Caudal et al, 2023). In general, the transcript level of every gene appears to be influenced by one or more eQTL.

In humans, the genetic basis of gene expression variation was dissected via GWAS on a large dataset of 49 tissues obtained for up to 838 individuals (Ferraro et al, 2020). This study clearly highlighted tissue-specific gene expression patterns and detected a large number of eQTL across tissues. The detected variants are mostly local eQTL (i.e., located close to the genes they influence), while distant eQTL (i.e., located far from the genes they influence) remains difficult to identify in such context due to the large genomes, limited sample size and low statistical power. As most of gene expression variation does not arise from local eQTL, part of the variance is therefore still unexplored. More recently, a population-scale transcriptomic analysis was performed on more than 1000 *Saccharomyces cerevisiae* yeast isolates leading to a deeper view of the genetic control of gene expression (Caudal et al, 2023). Overall, local eQTL were less frequent, representing 26% of the total set of eQTL detected, which is consistent with previous observation in a yeast cross (Albert et al, 2018). Nevertheless, the detected local and distant eQTL together only explained a small fraction of the gene expression heritability (Caudal et al, 2023), which is usually the case for all complex traits (Manolio et al, 2009; Hindorff et al, 2009). Several sources can potentially be considered as the origin of this missing heritability, i.e., the part of the phenotypic variance not unexplained by associated causal loci. These sources include the low power to detect small effect variants, to estimate non-additive effects (Cordell, 2009; Mackay, 2014; Zuk et al, 2012) but also the fact that rare and low-frequency variants (Manolio et al, 2009; Hindorff et al, 2009; Gibson, 2012; Pritchard, 2001; Walter et al, 2015) as well as structural variants

[1]Université de Strasbourg, CNRS, GMGM UMR 7156 Strasbourg, France. [2]Institut Universitaire de France (IUF), Paris, France. [3]These authors contributed equally: Andreas Tsouris, Gauthier Brach. ✉E-mail: jing.hou@unistra.fr; schacherer@unistra.fr

(Peter et al, 2018) are not systematically taken into account in GWAS.

Unlike common variants, rare and low-frequency variants are not adequately captured or tested in standard GWAS analyses, as they are only present in less than 1% or 5% of the population, respectively. In humans, these variants have been shown to play a role and alter adult height for example (Marouli et al, 2017; Akiyama et al, 2019). In addition, recent estimates of the impact of rare variants on the heritability for two human traits (height and body mass index) from whole-genome sequence data on 25,465 unrelated individuals of European ancestry confirmed that these variants are probably a major source of the missing heritability (Wainschtein et al, 2022). In the *S. cerevisiae* yeast model, the effect of rare and low-frequency variants on phenotypic variance has also been tested, since a bias towards this type of variants is observed, with more than 90% of the SNPs having a minor allele frequency (MAF) lower than 0.05 in a dataset of 1011 yeast genomes (Peter et al, 2018). Based on this resource, two independent surveys have shown that these variants contribute disproportionately to growth variation in a large number of conditions observed across yeast natural isolates (Bloom et al, 2019, Fournier et al, 2019). Although there is now strong evidence for the impact of such variants on organismal traits, their effect on molecular traits, such as gene expression variation, has been poorly studied at the population level.

Regarding the structural variants (SVs), they have long been recognized as an important source of genetic diversity and phenotypic variation in all organisms (Weischenfeldt et al, 2013). A few studies have provided catalogs of SVs in large populations of various organisms, including human, tomato, and yeast (Alonge et al, 2020; Li et al, 2023; O'Donnell et al, 2023; Liao et al, 2023), however, these catalogs are still non-exhaustive. Indeed, SVs are difficult to systematically and exhaustively detect for large natural populations and thus they are often excluded from complex trait association studies. In humans, certain SVs have been identified as involved in various diseases and are generally presumed to act through their effects on gene expression (Weischenfeldt et al, 2013). The impact of SVs on gene expression variation was estimated by exploring the effect of 61,668 SVs in 613 individuals on gene expression (Scott et al, 2021; Chiang et al, 2017). It was found that common SVs are causal for 2.66% of the eQTL and that SVs often affect multiple nearby genes (Scott et al, 2021). The impact of SVs on neighboring genes has also been recently studied in other organisms, such as tomato and yeast, and the results have shown variable effects on gene expression depending on the type of SVs (Alonge et al, 2020; O'Donnell et al, 2023). Although these studies represent the most comprehensive analysis of the impact of SVs on gene expression to date, they are still limited because either only part of the SVs are detected and not all SV-gene expression associations can be tested. Consequently, these studies do not provide a comprehensive view of the effect of SVs on the transcriptional landscape at the population level.

Understanding the effect of low-frequency variants as well as SVs on gene expression variation remains a challenge but it should provide deeper insights into the molecular basis of phenotypic diversity. Here, we took advantage of a diallel hybrid design in the *Saccharomyces cerevisiae* yeast model, which allows for exhaustive characterization of these two types of variants and their associations with gene expression variation in the population. Our findings show the need for a comprehensive catalog of genetic variants to identify the genetic basis of trait variation.

# Results

## Diallel design and transcript abundance variation

To better understand the sources of missing heritability of complex traits and more specifically of molecular traits such as transcript abundance, we sought to investigate the underlying genetic architecture of gene expression variation through the use of a diallel panel. In this context, a diallel cross panel was constructed by crossing a set of 26 genetically diverse *S. cerevisiae* natural isolates (Fig. 1, Dataset EV1). To capture a wide range of genetic diversity, different isolates from various ecological (e.g., beer brewery, fruit, clinical samples), and different geographical locations (e.g., Ireland, China, USA) were selected (Appendix Fig. S1A,B). The pairwise nucleotide divergence between parental isolates ranges from 0.03 to 1.1%, with an average of 0.59% (Dataset EV2). Parental isolates were crossed in pairs in all possible non-reciprocal combinations, a configuration often referred to as a non-reciprocal diallel cross. This diallel panel resulted in a total of 351 hybrids with 325 heterozygous hybrids coming from the cross of two different parents, and 26 homozygous hybrids resulting from the cross of genetically identical parental isolates.

To quantify gene expression variation in this population, RNA sequencing was performed on all the 351 hybrids (Tsouris et al, 2024). High-quality transcriptomes were obtained for 323 unique hybrids with over one million reads mapped to the reference genome. We obtained the expression levels (as transcript per million or tpm) for 6186 genes that are expressed in at least half of the samples (tpm > 0), including 5740 core ORFs as well as 422 accessory ORFs that are variably present in the generated hybrids (Dataset EV3). The expression abundance (as the mean of $\log2(tpm + 1)$) and dispersion (as the mean absolute deviation of $\log2(tpm + 1)$) of each gene in the diallel population correlates with those determined in a large population of 969 *S. cerevisiae* natural isolates ($R = 0.79$ and $R = 0.72$, respectively) (Appendix Fig. S1C,D) (Caudal et al, 2023). In addition, gene expression levels are also correlated between the homozygous hybrids generated in the current dataset and the original parental isolates that were characterized in the previously generated population-level dataset (Caudal et al, 2023) ($R = 0.73$) (Appendix Fig. S1E).

## Detection of the SNP-eQTL using the diallel panel

To determine the genotypes of all the hybrids, we retrieved the biallelic genetic variants of the 26 parental strains from the genomic sequences of the 1011 yeast genomes project (Peter et al, 2018). As a diallel panel leads to a highly structured population, it may introduce some biases when performing genome-wide associations. Singletons, i.e. genetic variants that are only present in a parental genetic background, show a strong linkage throughout the genome. We therefore removed these variants from the genotype matrix, resulting in a SNP matrix of 31,909 variants with a minor allele frequency (MAF) higher than 5%. Using this genotype matrix, we quantified the genome-wide heritability ($h^2_g$) across the 6186 gene expression traits (Appendix Fig. S2A, Dataset EV4). The median $h^2_g$

## Parental genotypes

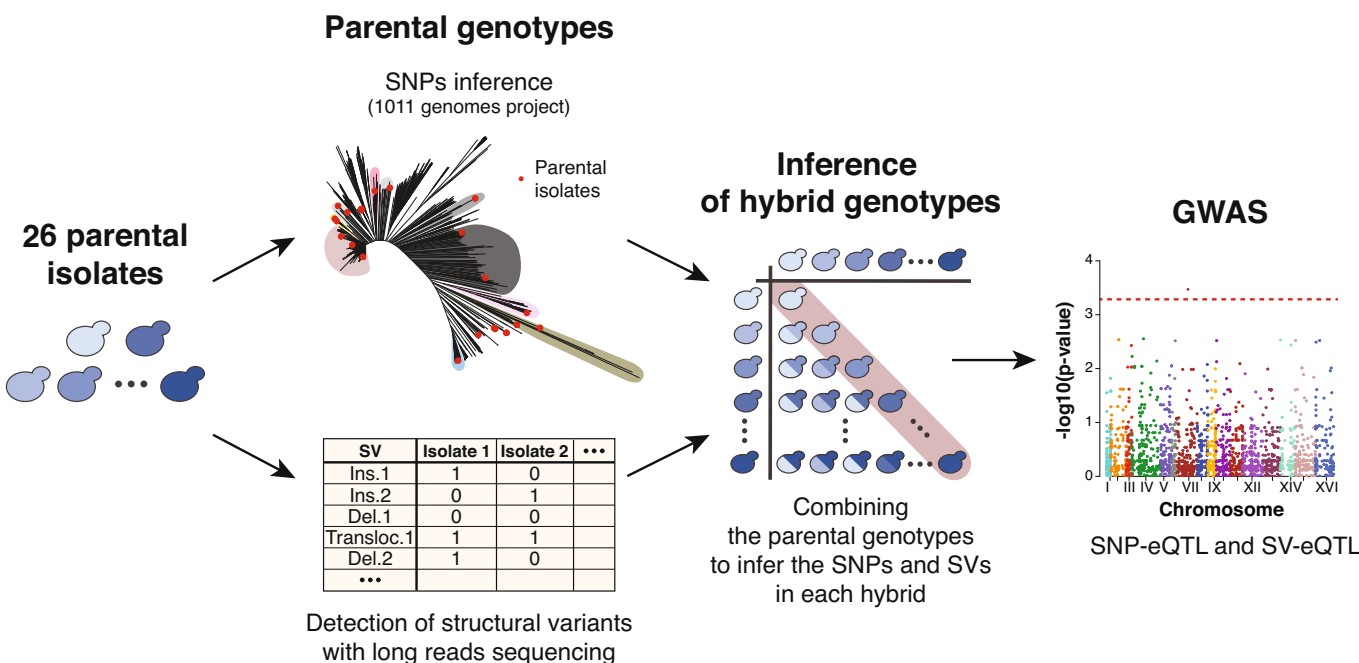

SNPs inference
(1011 genomes project)

Parental isolates

**26 parental isolates**

**Inference of hybrid genotypes**

**GWAS**

SV | Isolate 1 | Isolate 2 | •••

Combining the parental genotypes to infer the SNPs and SVs in each hybrid

Detection of structural variants with long reads sequencing

SNP-eQTL and SV-eQTL

**Figure 1. Workflow for genome-wide association using a diallel panel.**

A set of 26 natural isolates were selected as parents for a diallel panel. The SNPs in each isolate were inferred from the data of the 1011 yeast genomes project (Peter et al, 2018) while their SVs were detected using long-reads Oxford Nanopore sequencing. The parental genotypes were combined pairwise to infer the SNPs and SVs in each of the hybrids produced by the diallel panel. Finally, the SNP and SV genotypes of the hybrids were used to perform GWAS on the transcript abundance of 6186 genes.

is 0.28, which is similar to heritability of gene expression estimated in linkage mapping and GWAS in yeast (median = 0.26) as well as in other organisms such as in humans (mean = 0.03–0.26) (Albert et al, 2018; Zhang et al, 2022; Huan et al, 2015; Ouwens et al, 2020).

To identify SNPs that influence gene expression variation (SNP-eQTL), we performed GWAS with the transcript abundance traits of 6186 genes and using the SNP matrix as the genotype (Methods). We detected a total of 3039 SNP-eQTL that influence the level of transcript abundance of 1714 genes (Fig. 2A, Dataset EV5). Most SNP-eQTL impact only one or a few traits while few SNP-eQTL influence more than 20 phenotypes (Appendix Fig. S2B). In fact, we detected a total of 4 eQTL hotspots located in the *BCY1*, *GAL2* and *MLP1* genes, while another one is located in the promoter of *GSH2* and are associated to the expression of 83, 65, 62, 57 genes, respectively (Fig. 2B). The number of hotspots detected here is intermediate between the number identified by linkage mapping using single cross segregants and GWAS across a large set of natural isolates (Albert et al, 2018; Caudal et al, 2023). In addition, most traits are influenced by a single SNP-eQTL while very few are controlled by more than 5 (Appendix Fig. S2C). Overall, a trait was associated to 1.77 eQTL while each eQTL influenced 2.94 different phenotypes on average. A total of 1032 unique SNPs were associated with the 3039 eQTL (Dataset EV5).

We then distinguished the SNP-eQTL according to their relative position with respect to the gene they impact, with local eQTL being close to the affected gene and distant eQTL located further away or on different chromosomes (see Methods). We found 2952 distant and 36 local eQTL, while 51 eQTL involved HGT genes for which the precise locations of the genes were undetermined (Dataset EV5). Consistent with previous observations, local SNP-

eQTL display larger effect sizes compared to the distant ones (Wilcoxon test, *p*-value = 1.5e−4) (Fig. 2C) (Albert et al, 2018; Caudal et al, 2023).

Gene expression regulation of accessory genes has been shown to differ from that of the core genome (Caudal et al, 2023). We therefore compared the likelihood of accessory and core genes to be associated to SNP-eQTL (Fig. 2D). Accessory genes are more likely to be associated with SNP-eQTL (Fisher test, *p*-value = 3.7e−3). In total, 40.5% of the accessory genes (171 out of 422 accessory genes) considered in GWAS are associated with an eQTL, compared to a 26.7% of all genes (1543 out of 5740 genes), with a 1.52-fold enrichment. In addition, SNP-eQTL associated with accessory genes have a larger effect size on average than those associated with core genes (Wilcoxon test, *p*-value = 3.7e−6). Specifically, the effect size of distant SNP-eQTL controlling accessory genes is larger than those controlling core genes (Wilcoxon test, *p*-value = 3e−10). However, no significant difference was observed between the local SNP-eQTL associated with the expression of accessory and core genes (Wilcoxon test, *p*-values = 0.48) (Appendix Fig. S2D).

## Impact of the low-frequency SNPs on gene expression variation

Next, we sought to explore the contribution of rare and low-frequency genetic variants (i.e., with a MAF < 0.01 and <0.05, respectively) to the observed gene expression variation. In fact, genetic variants considered by GWAS must have a relatively high frequency in the population to be detectable, usually over 0.05 for relatively small datasets (Visscher et al, 2017). Rare and low-frequency variants are therefore excluded from association studies.

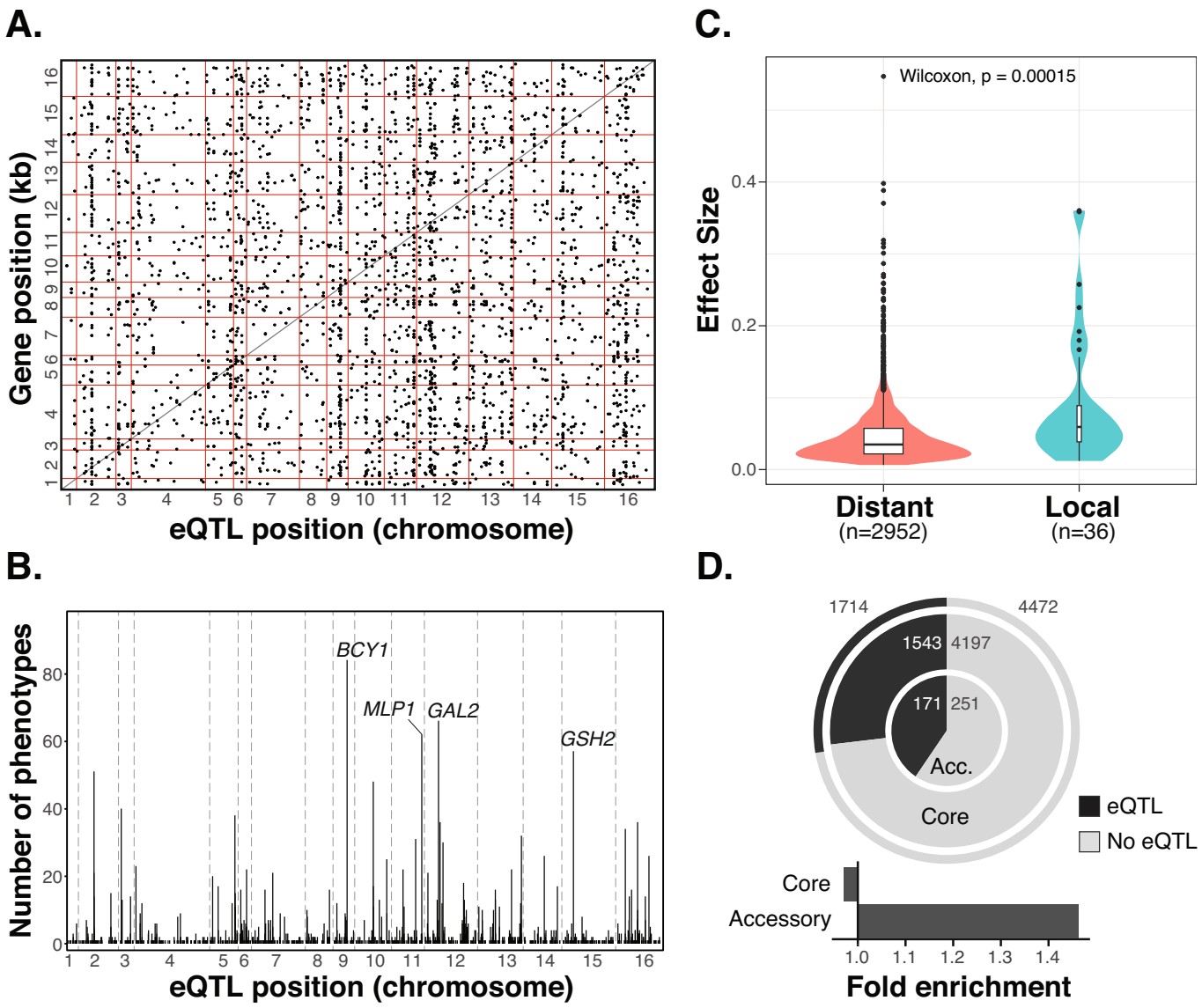

**Figure 2. Genome-wide association results using the SNP matrix.**

(A) Positions of the SNP-eQTL and their associated genes along the genome. Each point represents an eQTL and the horizontal and vertical red lines mark the margins of the chromosomes. The diagonal black line indicates the positions where the eQTL position and gene position are the identical, therefore eQTLs on that line are very close to the gene they influence. (B) Positions of SNP-eQTL and the number of genes they are associated with. (C) Comparison of the effect sizes of distant and local-eQTL inferred from GWAS (two-sided Mann-Whitney-Wilcoxon test, *p*-value = 1.5e−4). For distant-eQTL effect sizes, min = 0.006, 1st quartile = 0.022, median = 0.035, 3rd quartile = 0.057, max = 0.546. For local-eQTL effect sizes, min = 0.012, 1st quartile = 0.038, median = 0.059, 3rd quartile = 0.088, max = 0.36. (D) The number and proportions of core (outer circle) and accessory genes (inner circle) associated with eQTLs. Fold enrichment of core and accessory genes associated to eQTLs. Significance was calculated using a two-sided Fisher test.

However, our diallel panel provides a powerful strategy to assess the impact of low-frequency variants that are initially present in the population, i.e., in the 1011 *S. cerevisiae* collection here (Peter et al, 2018). In fact, each parental genotype is present several times, increasing the frequency of these variants in this new population and allowing their detection in GWAS. While the parental genomes carry 1228 low-frequency variants (48 of which are rare variants) as defined in the population of 1011 isolates, none of the genetic variants has a MAF lower than 0.058 in our diallel panel (Online Datafile 1).

Interestingly, the MAF of the genetic variants identified as SNP-eQTL is much lower than that of the entire set of SNPs considered for GWAS, suggesting that low-frequency variants have a large impact on transcript abundance variation (Fig. 3A). In fact, 8.4% (104 out of 1228) of the significant SNP-eQTL have a MAF below 0.05, whereas only 3.0% (928 out of 30,681) of SNP matrix used for GWAS have a MAF below 0.05 in the natural population (Fig. 3B). This nearly threefold enrichment in low-frequency SNPs in the set of associated variants is consistent with previous studies focusing on the influence of low-frequency variants on yeast growth traits

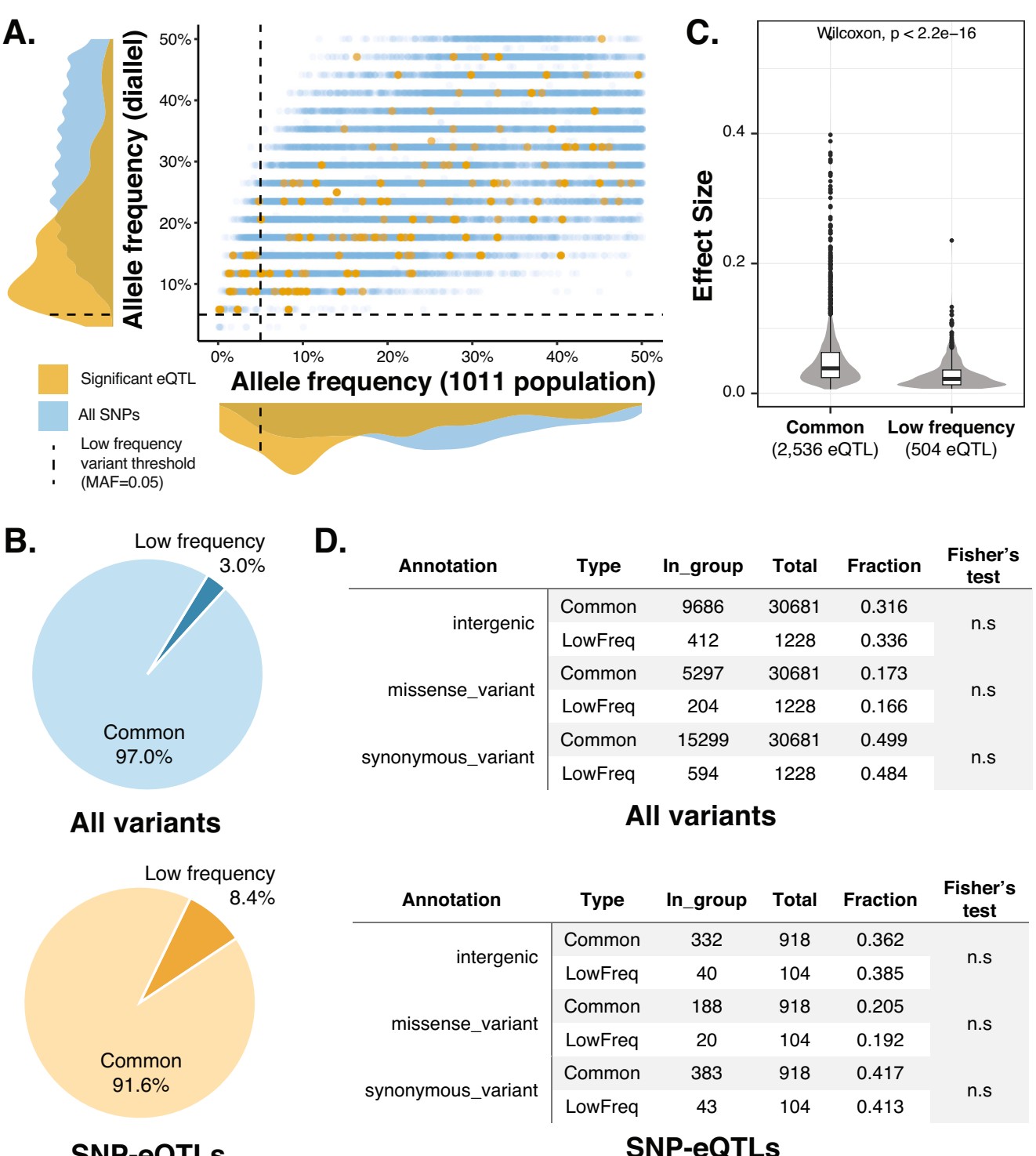

(Bloom et al, 2019; Fournier et al, 2019). Using a diallel panel, we previously have found that 16.8% of growth SNP-QTL were low-frequency variants in the collection of 1011 *S. cerevisiae* isolates (Fournier et al, 2019). Finally, we measured the respective effect size of each SNP-eQTL and we found that even if enrichment is observed, low-frequency variants have a lower effect size compared to the common variants (Fig. 3C).

To further characterize the functional effects of the associated variants, we annotated all variants in the SNP matrix using SnpEff (Cingolani et al, 2012). The majority of variants falls into three annotation groups, namely intergenic (10,098/31,909), synonymous_variant (15,893/31,909) and missense_variant (5501/31,909) (Dataset EV6). The remaining 417 variants mainly belonged to intron related variants and some rare cases of loss of start codon

**Figure 3.  Low frequency variants and gene expression variation.**

(A) Comparison of the MAF of the SNPs used in GWAS in the diallel population and the natural population (1011 isolates). Orange points represent significant eQTL SNPs whereas blue points represent the remaining SNPs. In the distributions along the axes, the orange distribution covers the significant SNPs, and the blue distribution covers all the SNPs used in GWAS. Vertical and horizontal dashed lines show the threshold of MAF (5%) used to define low-frequency variants. (B) Comparison of the fraction of low frequency variants in all the SNPs used in GWAS (blue) and the significant eQTLs (orange). (C) Comparison of the effect size of common and low-frequency SNP-eQTL. The number of associated variants (n) in each category are indicated. For common variants associated eQTL, the effect sizes show $min = 0.006$, 1st quartile $= 0.024$, median $= 0.039$, 3rd quartile $= 0.063$, $max = 0.547$. For low frequency variants associated eQTL, the effect sizes show $min = 0.007$, $1^{st}$ quartile $= 0.013$, median $= 0.022$, 3rd quartile $= 0.036$, $max = 0.235$. (D) Comparison of functional annotations between low frequency and common variants across the full SNP matrix (upper panel) and across the associated SNPs (lower panel). Fisher's exact tests were performed for all common vs. low frequency for each annotation group.

(start_loss, 13 variants) and gain of premature stop codon (stop_gained, 37 variants) (Dataset EV6). Using these annotations, we analyzed the distribution of different variant groups across common vs. low frequency variants (Dataset EV6). We observed no significant differences between common and low frequency variants across all annotation groups (Dataset EV6, Fig. 3D), indicating that low frequency variants are a random sampling of all SNPs. We performed the same comparison for all eQTL associated SNPs and observed the same trend (Dataset EV7, Fig. 3D). While there is no difference in terms of functional annotation between common and low frequency variants, we do observe that missense variants and intergenic variants are more likely to be associated with an eQTL than synonymous variants. Specifically, missense variants are 1.18-fold overrepresented in eQTL associated SNPs (Fisher's test P-value = 0.03), and intergenic variants are 1.15-fold overrepresented in eQTL associated SNPs (Fisher's test P-value = 0.02) compare to the full variant matrix.

## Characterization of the structural variants in the diallel panel

We then focused on the impact of SVs on gene expression variation using our diallel panel. Accurate and exhaustive identification of SVs in a large population can be very laborious and, in some cases, practically impossible. These variants are therefore only very rarely taken into account in association studies. However, the structure of a yeast diallel panel makes this task much easier because the SVs present in the hybrids correspond to the combination of the SVs present in their parents. Long-read genome sequencing and SV detection thus provide an exhaustive list of the SVs present in each of the 323 hybrids.

We therefore sequenced the genomes of 24 out of 26 parental isolates using a long-read Oxford Nanopore sequencing strategy, as two of the parental strains, S288C and Σ1278b, were already completely sequenced and assembled (Goffeau et al, 1996; Dowell et al, 2010). On average, we obtained 108,351 reads with a length of 12.8 kb for each parental genome (Dataset EV8). The mean coverage was 66X on average, ranging from 19.7X to 150X for the ABA and ACS isolates, respectively. We then used these long reads to generate de novo assemblies for the entire set of strains (see Methods). Overall, we obtained contiguous assemblies with a N50 of 835 kb and an average genome size of 12.2 Mb (Dataset EV9). In most cases, a single contig covered more than 90% of a chromosome, while in other cases, only a few contigs were needed. (Appendix Fig. S3A). The assembly of each parent was then compared to the genome of the reference strain (S288C) using MUM&co to detect all the SVs present in the parental genomes (O'Donnell and Fischer, 2020). We define SVs as variants that are

at least 50 bp in size. As the SVs are detected independently in each parent, we also had to identify the presence of the same event in the several genetic backgrounds in order to merge them into the same occurrence. For this purpose, we used Jasmine (Kirsche et al, 2023), a tool that compares the genomic position as well as the sequences of SVs, and we found a total of 1953 SVs in our set of parental genomes (Dataset EV10).

The SVs were classified into 6 types, namely insertions ($n = 1032$ variants), deletions ($n = 543$), duplications ($n = 250$), contractions ($n = 65$), translocation ($n = 33$) and inversions ($n = 30$) (Fig. 4A). The number of SVs in each parent correlates with the nucleotide divergence between that parent and the reference genome ($R = 0.71$, $p$-value $= 7.30e^{-05}$) (Appendix Fig. S3B). While Σ1278b has the lowest nucleotide diversity compared to S288C and the lowest number of SVs ($n = 153$), the BAM isolate has the highest number of SVs (n = 357) and corresponds to the most divergent strain with respect to S288C (Fig. 4B). On average, we detected 241 SVs per genome. The distribution of SV types in each parent is similar to their overall abundance. Most SVs are insertions and deletions, while the remaining SVs types only represent a small fraction. As with SNPs, SVs exhibit a bias towards low-frequency alleles (Appendix Fig. S3C). A total of 60.1% of SVs ($n = 1190$) are low-frequency variants, with only 763 SVs have a MAF greater than 0.05 (Appendix Fig. S3C).

Finally, we also identified SVs that are related to transposable elements, a major source of structural variation (see Methods). Overall, 869 SVs are Ty-related (44.5%), while 1084 (55.5%) are not (Appendix Fig. S3D). Ty-related SVs can be separated into two groups according to their length. The first group of SVs are about 340 bp in length and contains the Ty Long Terminal Repeats (LTR), while the second group of SVs are about 6 kb in length and contains the entire or partial Ty elements (Appendix Fig. S3D). A total of 64.2% ($n = 558$) of Ty related SVs only contain LTR sequences, while the remaining 35.8% ($n = 311$) contain partial or complete Ty elements.

## Impact of SVs on gene expression phenotypes

Before performing genome-wide associations, we inferred the genotypes of the 323 hybrids by combining the SVs of their parents as it was done previously for the SNPs. We excluded the singletons, i.e., all variants present in only one of the 26 parents, because they would cause many false positive associations due to severe linkage throughout the genome. As a result, we obtained a genotype matrix of 763 SVs across the 323 hybrids. The $h^2_g$ of transcript abundance traits using the SV genotype matrix was 0.021, an order of magnitude lower than the $h^2_g$ obtained with the SNP matrix ($h^2_g = 0.28$) (Appendix Fig. S4A).

## A.

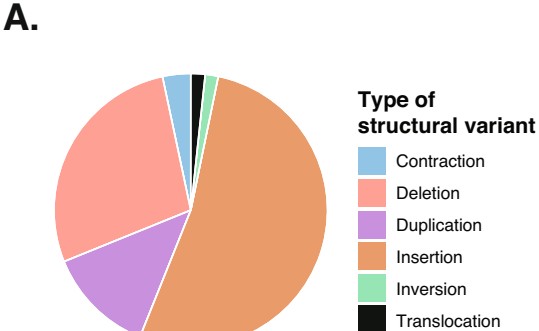

**Type of structural variant**
- Contraction
- Deletion
- Duplication
- Insertion
- Inversion
- Translocation

## B.

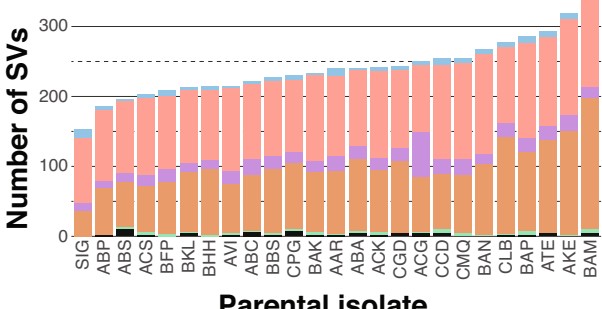

## C.

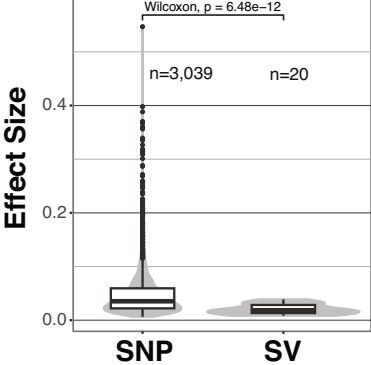

**Figure 4. Structural variants detected in the parental genomes and effect size comparison.**

(A) Fractions of the different types of structural variants identified in the parental strains. (B) Number of the SVs detected in each parental strain. Each parent is pictured in each stacked bar and colors of the bars represent the different types of SV as above. (C) Comparison of the effect sizes of SNP-eQTL and SV-eQTL identified through GWAS (two-sided Mann-Whitney-Wilcoxon test, $p$-value = 6.48e−12). The sample size n corresponds to the number of SNP- and SV-eQTL and are indicated on the plot.

To identify SVs that influence gene expression variation (SV-eQTL), we performed GWAS with the transcript abundance traits of 6186 genes and using the SV matrix as the genotype (see Methods). We only detected 20 significant SV-eQTL that influence the expression of 13 genes (Dataset EV11). A total of 19 unique SVs were associated. The transcript abundance of 10 genes was

influenced by one SV-eQTL, while the expression of two genes was influenced by multiple SV-eQTL (Appendix Fig. S4B). All SV-eQTL are associated to a single gene except for one that was associated to the transcript abundance of two genes (Appendix Fig. S4C). Most SV-eQTL are insertions and deletions ($n = 13$ and $n = 4$, respectively), while one SV-eQTL is a duplication located on chromosome 13 and another one a contraction on chromosome 15. SV-eQTL involving deletions show higher effect sizes than insertions, contraction and duplications, (Appendix Fig. S4D). Half of the SV-eQTL are *Ty*-related (10 out of the 20 SV-eQTL) but this is not significant compared to the overall abundance of *Ty*-related SVs (fisher test, $p$-value = 1). Overall, *Ty*-related SV-eQTL tend to show lower effect sizes compared to non-*Ty* SV-eQTL for the same variant type (Appendix Fig. S4D). Finally, since we assessed the relationship of SNPs and SVs to the same phenotypes across the same population, it was possible to compare the effect of SNPs and SVs on gene expression. In fact, we found that the effect sizes of SV-eQTL are overall smaller than those of SNP-eQTL (Appendix Fig. 4C).

## Discussion

Understanding the genetic basis of gene expression variation is essential to have a better insight into the genotype-phenotype relationship. While large-scale studies detected thousands of genetic variants involved in gene expression variation via genome-wide associations, these variants explain only a small fraction of gene expression variation (Ferraro et al, 2020; GTEx Consortium, 2017). In this context, we sought to investigate the impact of potential sources of this so-called missing heritability. Using a diallel design, we were able to assess the influence of low-frequency as well as structural variants on the variation in transcript abundance. On the one hand, this strategy makes it possible to artificially increase the frequency of genetic variants in a diallel panel and thus to take low-frequency variants into account in association studies. On the other hand, it is possible to easily establish the exhaustive list of SVs present in the hybrids from the parental genotypes. Based on a generated yeast diallel panel composed of 323 hybrids and associated with the corresponding transcriptomes, we carried out a GWAS for each gene expression trait and identified 3,039 SNP-eQTL and 20 SV-eQTL in total.

Regarding the low-frequency and rare variants, we found a strong enrichment of these variants in the set of significantly associated SNPs, i.e., SNP-eQTL. While only 3.0% of the SNPs considered in the association tests have a low frequency in the natural yeast population, we found that 8.4% of all SNPs-eQTL are low-frequency variants, showing a ~3-fold enrichment. These results clearly show the same trend and enrichment already observed for yeast growth phenotypes (Bloom et al, 2019; Fournier et al, 2019). This observation highlights that low-frequency variants strongly contribute to gene expression variation in a natural population. This is very important as much of the detected genetic polymorphisms in a population, such as the 1011 yeast genomes dataset, are low-frequency variants with almost 92.7% of the polymorphic sites associated with a MAF lower than 0.05 (Peter et al, 2018). Moreover, low frequency variants do not differ from common variants in terms of functional annotations, indicating that they are a random sample of all variants. Interestingly, low

frequency variants are consistently enriched for both QTL related to growth phenotypes and gene expression traits. Alongside this enrichment, we also found that overall, the effect sizes of these variants are smaller than those of the common variants for gene expression traits, contrasting to the growth phenotypes (Bloom et al, 2019; Fournier et al, 2019). This difference could possibly be due to the rich and lenient growth condition used for the gene expression measurements, contrasting to various stress conditions used for growth phenotypes. Further transcriptomic analyses across different culture condition are still needed to explore more deeply the effects of low-frequency variants in a population.

Our diallel panel, however, showed some limitations in exploring the impact of SVs on gene expression variation. In fact, we could only detect a total of 19 SV-eQTL probably because very few SVs were taken into account compared to SNPs in the genotype matrices (763 SVs versus 31,909 SNPs). By taking into account the size of the genotypic matrices, we can observe that the same fraction of SNP-eQTL and SV-QTL was globally detected, corresponding respectively to 3.2% and 2.5% of associated genetic variants, respectively. It is clear that SVs are widely present in natural populations, as in *S. cerevisiae* for example. A recent study of 142 telomere-to-telomere genomes from natural isolates characterized approximately 4800 unique SVs and identified 97 SVs that influence the expression of neighboring genes (O'Donnell et al, 2023). The impact of SVs on gene expression has also been shown in humans where SVs are responsible for 2.66% of eQTL and they often affect multiple nearby genes (Scott et al, 2021). The impact of SVs is not limited to gene expression traits, a survey of over 100 tomato genomes has indeed identified hundreds of SV-QTLs impacting several volatile flavor and fruit metabolites (Alonge et al, 2020; Li et al, 2023).

Our diallel design also comes with other limitations. First, the diallel hybrid population is very structured, and in order to avoid biases we had to eliminate many variants that were in linkage, therefore limiting the number of variants used in GWAS. This limit could possibly be circumvented by significantly increasing the size of the panel. Second, the hybrids were not sequenced directly but their genotypes were inferred from their parents. Therefore, any potential genomic modifications that have arisen since the formation of the hybrids were generated (such as any mutation, loss of heterozygosity or other structural events) are not considered in our design. Moreover, as the mitochondrial genomes in the hybrid cannot be easily inferred as the nuclear genome due to potential recombination between the parental types, the contribution of the mitochondrial genome on gene expression is not considered here. Finally, as already mentioned, the number of SVs used for the association studies was probably insufficient to have a global view of the impact of SVs on gene expression.

A population-scale comprehensive survey would now be essential to have an accurate and exhaustive view of the impact of SVs on gene expression variation. The collection of 1011 natural yeast isolates could be an excellent resource for such exploration. Long-read sequencing of this large collection would lead to a species-wide view of the structural variants as well as a comprehensive pangenome graph (Liao et al, 2023; Hickey et al, 2023). The exhaustive catalog of SVs generated could then be used to perform GWAS on a large number of traits. Many traits have already been measured for this large collection of *S. cerevisiae* isolates, including growth phenotypes, as well as molecular traits like transcript and protein abundance (Caudal et al, 2023; Peter et al, 2018; Muenzner et al, 2022; Teyssonniere et al, 2024). A better view of the impact of SVs on traits would definitely improve our understanding of the genotype-phenotype relationship by revealing their real role in missing heritability.

# Methods

## Diallel panel generation and cell harvesting

The hybrid diallel panel was produced from a series of pairwise crosses between 26 haploid parental strains. Two stable haploid lines were generated for each parental strain, the *MAT*a lines carry a *KanMX* cassette and the *MAT*alpha lines carry a *NatMX* cassette, each cassette replacing the *HO* locus (Fournier et al, 2019). For each cross cells of opposite mating type were patched together on solid YPD media (1% yeast extract, 2% peptone, and 2% glucose) and incubating them at 30 °C overnight. To select for hybrid cells by selecting the combination of both cassettes the cell patches were then transferred to YPD media containing G418 (200 mg/ml) and nourseothricin (100 mg/ml) and incubated at 30 °C overnight. All procedures were done using the replicating robot ROTOR (Singer Instruments). In total, we obtained 351 genetically unique hybrids.

Hybrids were then incubated in liquid synthetic complete (SC) media with 2% glucose in 96-deep-well plates for cell harvesting. The optical density of each hybrid's culture was measured systematically using a 96-well microplate reader (Tecan Infinite F200 Pro). During log-phase growth ($OD_{600nm} \sim 0.3$), cells were harvested by filtration in 96-well filter plate (Norgen, #40008) where the media was eliminated by applying vacuum (VWR, #16003-836). The filter plates were immediately flash frozen in liquid nitrogen and stored at −80 °C.

## cDNA library preparation

We used the Dynabeads® mRNA Direct Kit (ThermoFisher #61012) to extract the mRNA of each hybrid. The cells were lysed using glass beads and were then incubated at 65 °C for 2 min. mRNA was selected with two rounds of hybridization of their polyA tails to magnetic beads coupled to oligo(dT) residues.

cDNA sequencing libraries were prepared with the NEBNext® Ultra™ II Directional RNA Library Prep Kit (NEB, #E7765L) and using the manufacturer's protocol. The concentration of cDNA in each library was quantified using the Qubit ™ dsDNA HS Assay Kit (Invitrogen ™) in a 96-well plate using a microplate reader (Tecan Infinite F200 Pro) with an excitation frequency of 485 nm and emission of 528 nm. Fragment size was assessed with Bioanalyzer 2100 (Agilent™) using the High sensitivity DNA kit (#5067-4626). We generated sequencing pools containing equimolar fragments from each sample. Lastly, the pools were sequenced for 75 bp single-end with Nextseq 550 (Illumina™) sequencer at the EMBL Genomics Core Facility.

## mRNA abundance quantification

For each sample the raw sequencing reads were mapped to a custom reference genome using STAR (Dobin et al, 2013) with the following parameters:

```
--outSAMtype BAM SortedByCoordinate \
--outFilterType BySJout \
--outFilterMultimapNmax 20 \
--outFilterMismatchNmax 4 \
--alignIntronMin 20 \
--alignIntronMax 2000 \
--alignSJoverhangMin 8 \
--alignSJDBoverhangMin 1
```

The custom reference genome combines the 16 chromosomes of the R64_nucl reference genome with the accessory ORFs present in the parental strains ($n = 665$) according to the data from the 1011 yeast genomes project (Peter et al, 2018), each accessory ORF is considered as a separate contig. In total, 323 hybrids had enough reads and were used in the subsequent analyses. The reads aligning to each gene of the reference ($n = 6285$) and accessory ($n = 665$) genomes were counted using the featureCounts function of the R package subread (Liao et al, 2014) with the parameter *countMultiMappingReads = F* in order to not take multi-mapped reads into account. For a given hybrid, if accessory genes that have orthologs in the reference genome were annotated as absent, we merged their reads counts to those of their reference genome counterparts. mRNA abundance was then normalized by calculating the transcripts per million value (tpm) of each gene. This gave us a list of tpm values for 6917 genes. We filtered out genes that have a zero tpm value in more than half of all samples, leading to a final dataset of 6186 genes (Tsouris et al, 2024).

## Functional annotation of the SNP variants

SnpEff (v5.1) (Cingolani et al, 2012) was used to annotate variants located in coding regions in the vcf file (reference genome R64-1-1.105). The annotation file in .vcf format can be found in Datafile 2.

## Long-reads sequencing of the parental genomes

To carry out long-reads sequencing of the parental isolates with Oxford Nanopores Technologies (ONT) sequencing, we first incubated each parental isolate in 40 ml of YPD (1% yeast extract, 2% peptone and 2% glucose) at 30 °C for 48 h. The cells were then harvested by centrifugation at $7000 \times g$ for 3 min and then resuspended water before a second round of centrifugation. The cell pellet was resuspended in 4 mL of sorbitol 1 M containing 250 µl of zymolyase (1000 U/ml) and incubated at 30 °C for 2 h with agitation. Protoplasts were recovered by centrifugating for 5 min at $2000 \times g$ at 4 °C and discarding the supernatant. They were then resuspended in 4 mL of lysis buffer, a solution of Tris-HCl 125 mM, EDTA 62.5 mM, NaCl 0.625 M, 0.05 g PVP40, 1.25% SDS and 10 mg/ml RNaseA. The cell lysis suspension was incubated at 50 °C for 3 h and was then chilled on ice for 2 min. We then added 5 ml TE and 3 ml AcK 5 M to the suspension and centrifugated twice at $2000 \times g$ for 15 min at 4 °C. After each centrifugation the supernatant was transferred to a new Falcon tube where we then added 12 ml of isopropanol to precipitate the genomic DNA. After a 5-min centrifugation at $500 \times g$ the DNA pellet was washed with ice cold 70% ethanol and incubated for 5 min on ice. To precipitate the DNA pellet, we carried out a centrifugation of 5 min at $500 \times g$ at 4 °C and then removed all ethanol from the tube. Finally, we added 250 µl of TE without disturbing the pellet and incubated overnight at room temperature

before recovering the resuspended DNA without recovering any of the remaining pellet.

Some parental isolates were sequenced in-house using MinION R9.4.1 flowcells (10 isolates) whereas 14 isolates where sequenced by the Genoscope using PromethION R9.4.1 flowcells. All libraries were prepared with the Ligation Sequencing Kit SQK-LSK109. In order to sequence multiple isolates in the same run, the isolates sequenced with the MinION and PromethION were barcoded with the EXP-NBD104 and NBD114.96 Native Barcoding Kits respectively. Basecalling was carried out with guppy 5.0.16 (MinION data) and 5.0.11 (PromethION data).

## Genome assembly and structural variant detection

Raw fastq files were processed using Porechop (v.0.2.4; github.com/rrwick/Porechop) in order to remove adapters and barcodes. We further treated these reads using Filtlong (v.0.2.1; github.com/rrwick/Filtlong) to remove reads shorter than 1000 nucleotides. For each strain, reads were then assembled using both Canu (v. 2.2) (Koren et al, 2017) and SMARTdenovo (Liu et al, 2021). Canu assemblies were generated using the options *-nanopore-raw* and *genomeSize = 12 m* while we used the default options for SMARTdenovo. We used the same fastq reads to polish both assemblies per strain using Medaka (v. 1.7.2; github.com/nanoporetech/medaka).

We then set out to detect SVs using MUM&Co (v 3.8) (O'Donnell and Fischer, 2020) separately on each genome assembly, using the option *-g 12000000*. Unique identifiers were also added to each SV using a combination of bcftools (Danecek et al, 2021) and the R package vcfR (v1.12) (Knaus and Grünwald, 2017). We then used Jasmine (Alonge et al, 2020; Kirsche et al, 2023) (to collapse SV calls during two rounds. First, for each strain, SV calls from both Canu and SMARTdenovo assemblies were collapsed into 24 strains specific VCF files. During the second round of merging, the 24 VCF files and a 25th containing direct SV calls from the publicly available Σ1278b genome (Dowell et al, 2010) were collapsed into one final VCF file for downstream analyses. Then, since the 26th parent is a haploid version of the reference strain we assigned all the SVs as absent in that parent. Finally, SVs linked to transposable elements were detected using BLAST (Johnson et al, 2008) and a database containing LTR and Ty sequences (Bleykasten-Grosshans et al, 2021). Any SV covered by Ty-related elements on more than 50% of its length was flagged as Ty-related.

## Generation of the genotype matrix

We recovered the genotypes for all the hybrids in our diallel panel from Fournier et al, 2019 where the parental genotype had been combined to infer the genotypes of the hybrids (Fournier et al, 2019). We retained biallelic variants using vcftools with the option *--min-alleles 2* and excluded singletons with the vcftools *--singleton* and *--exclude* commands, resulting in a final matrix of 31,818 SNPs. The genotype matrix was then recoded to a bed file with the 'recode12' function of PLINK (Chang et al, 2015). The SVs were detected using MUM&co were subjected to the same filtering and recoding process, in order to be used for GWA analyses. The gene expression phenotypes, measured as transcripts per million, of 6186 genes were recovered from Tsouris et al, 2024. The z-scores of each phenotype were then calculated and later used in GWAS.

## Genome-wide heritability estimation

The genotype matrices used to estimate the genome-wide heritability were generated from the genotype data of the 1011 yeast genomes project (for the SNP matrix) and from the SVs identified using long-reads sequencing (for the SV matrix). To decrease linkage between the individuals we removed the variants present in only one parent. Furthermore, the variants with strong linkage disequilibrium ($r^2 > 0.8$) were removed for the calculation of the genome-wide heritability estimation. We calculated the weights of each variant using ldak with the –cut-weights and –calc-weights-all arguments and the default parameters (Zhang et al, 2021). All variants with non-zero weights were to generate a filtered vcf matrix of 5493 SNPs that was then recoded with the plink -make-bed command. The filtered and recoded matrix was used to calculate the kinship between the individuals using the popkin function of the R package popkin with the default parameters. To estimate genome-wide heritability ($h^2_g$), from the kinship matrix mentioned above we used the hglm R package (Rönnegård et al, 2010) using the default parameters (Tsouris et al, 2024).

## Genome-wide association

Genome-wide association analyses on transcript abundance z-scores were performed with the single_snp function of the FaST-LMM python package (Widmer et al, 2014). We calculated condition-specific p-values by permuting the phenotypic values of the individuals 100 times and setting average 5% quantile (5% lowest p-values) as the threshold. This method doesn't provide us with a false discovery rate (FDR). Independent GWA analyses were carried out for SNPs and SVs but the p-value thresholds for the GWA on SNPs and on SVs were very different due to the important difference between the number of SNP and SV used. To correct for this discrepancy, we adjusted the condition-specific p-values of each GWA by normalizing to the total number of variants (SNPs and SVs).

$$adjThresh_{SNPs} = \frac{thresh_{SNPs} \cdot SNPs}{SNPs + SVs} \text{ and }$$
$$adjThresh_{SVs} = \frac{thresh_{SVs} \cdot SVs}{SNPs + SVs}$$

The GWA eQTL displayed some genetic linkage associated patterns/signatures driving groups of linked variants to pass the significance threshold. We calculated genetic linkage between variants using the plink --r command (Chang et al, 2015) and for each linkage group with an R value over 0.8 we only retained the variant with the lowest p-value. This variant filtering steps were independently run for the GWASs using the SNP and SV.

To distinguish between local and distant eQTLs we used a threshold of 25 kb, where the variants withing 25 kb of the start site of a gene were considered as local whereas those further away were considered as distant. We tested the differences in effect sizes or absolute variant weight between groups of eQTLs with Wilcoxon-Mann-Whitney tests using *wilcox.test* R function with the default parameters.

## Analysis of low frequency variants

The MAF of the SNPs in the hybrids and the natural population of 1011 isolates was calculated with the vcfR using the *maf* command (Knaus and Grünwald, 2017) using the genotype matrix that was used for GWAS and the genotype matrix of the 1011 yeast genomes project respectively. The fold enrichment values of eQTLs from low frequency and common variants were carried out manually in R and tested using the *fisher.test* command.

## Data availability

All Oxford Nanopore sequencing reads are available in the European Nucleotide Archive (ENA) under the accession number PRJEB64478. The 1002 Yeast Genome website - http://1002genomes.u-strasbg.fr/files/ diallel_RNAseq/GWAS provides access to: - Datafile 1: MAF of the SNPs across the diallel population and the population of 969 natural isolates. - Datafile 2: Functional annotation of all SNP variants.

## Peer review information

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

## Acknowledgements

This work was supported by a National Institutes of Health (NIH) grant R01 (GM147040-01), a European Research Council (ERC) Consolidator grant (772505) to JS and a French National Research Agency (ANR) young investigator grant (ANR-22-CE12-0023-01) to JH. It is also part of Interdisciplinary Thematic Institutes (ITI) Integrative Molecular and Cellular Biology (IMCBio), as part of the ITI 2021-to-2028 program of the University of Strasbourg, CNRS, and Inserm, supported by IdEx Unistra (ANR-10-IDEX-0002). JS is a Fellow of the University of Strasbourg Institute for Advanced Study (USIAS) and a member of the Institut Universitaire de France.

## Author contributions

**Andreas Tsouris**: Conceptualization; Formal analysis; Investigation; Methodology; Writing—original draft. **Gauthier Brach**: Formal analysis; Investigation; Methodology. **Anne Friedrich**: Formal analysis; Methodology. **Jing Hou**: Conceptualization; Formal analysis; Funding acquisition; Investigation; Methodology; Writing—review and editing. **Joseph Schacherer**: Conceptualization; Supervision; Funding acquisition; Investigation; Methodology; Writing—original draft; Writing—review and editing.

## Disclosure and competing interests statement

The authors declare no competing interests.

