## [Peer Review File · Molecular Systems Biology]

Diallel panel reveals a significant impact of low-frequency variants on gene expression variation

Andreas Tsouris, Gauthier Brach, Anne Friedrich, Jing Hou, and Joseph Schacherer

Corresponding author(s): Joseph Schacherer (schacherer@unistra.fr) , Jing Hou (jing.hou@unistra.fr)

Review Timeline:

Submission Date:	15th Sep 23
Editorial Decision:	11th Oct 23
Revision Received:	18th Dec 23
Editorial Decision:	24th Jan 24
Revision Received:	29th Jan 24
Accepted:	30th Jan 24

Editor: Maria Polychronidou

Transaction Report:

11th Oct 2023

Manuscript Number: MSB-2023-12009

Title: Diallel panel reveals a significant impact of low-frequency variants on gene expression variation

Dear Joseph,

Thank you again for submitting your work to Molecular Systems Biology. We have now heard back from the three reviewers who agreed to evaluate your study. Overall, the reviewers acknowledge that the presented findings are potentially interesting. However, they do raise a series of substantial concerns regarding the conclusiveness and impact of the study. Notably, reviewer #3 is not convinced that the study presents a significant advance over existing knowledge in the field. During our cross-commenting process, in which the reviewers get the chance to make additional comments based on each other's reports, reviewers #1 and #2, mentioned that in their opinion the study is a potentially valuable contribution to the field, pending substantial revisions. Reviewer #3 mentioned that while they are still not convinced that the advance is major, they would see value in extending the study and providing further insights into the effect of the low frequency variants (e.g. on whether these low-frequency variants have any particular features). On balance, given that the reviewers did have positive words about the potential relevance of the study, we have decided to offer you the chance to address the issues raised in a major revision.

Without repeating all the points listed below, some of the more fundamental issues are the following:

- The broader relevance of the main findings and conclusions for a general audience, beyond yeast, needs to be better supported. It is important to clarify the main contribution and significance of the study for quantitative genetics.
- Further insights into the observed effect of the low frequency variants would be required to enhance the impact of the study.
- All three reviewers raise several technical issues that need to be addressed in order to better support the main conclusions.

All issues raised by the reviewers need to be satisfactorily addressed. The reviewers make several constructive suggestions on how to address the issues raised. I have also included below the additional comments of the three reviewers from referee cross-commenting, as they provide helpful guidance on how to improve the study. We recognize that the requested revisions are substantial. As you may already know, our editorial policy allows in principle a single round of major revision, so it is essential to provide responses to the reviewers' comments that are as complete as possible. Please feel free to contact me in case you would like to discuss in further detail any of the issues raised or if you would like to share your revision plan with me. I would be happy to schedule a call.

On a more editorial level, we would ask you to address the following points:

- The keywords need to be reduced to 5.
- Please provide a .doc version of the manuscript text (including legends for the main figures) and individual production quality figure files for the main Figures (one file per figure).
- We have replaced Supplementary Information by the Expanded View (EV format). In this case, all additional figures can be included in a PDF called Appendix. Appendix figures should be labeled and called out as: "Appendix Figure S1, Appendix Figure S2... Appendix Table S1..." etc. Each legend should be below the corresponding Figure/Table in the Appendix. Please include a Table of Contents in the beginning of the Appendix. For detailed instructions regarding expanded view please refer to our Author Guidelines: .
- Tables S1-S9 and Datafile 1 should be provided as EV Datasets (either as .xls files or .zip folders). Please provide one file per EV Dataset. Please include the description of each EV Dataset in the dataset file itself, ie. in a separate tab for .xls files or as a README.txt file in .zip folders.
- Please provide a "standfirst text" summarizing the study in one or two sentences (approximately 250 characters), three to four "bullet points" highlighting the main findings and a "synopsis image" (550px width and max 400px height, jpeg format) to highlight the paper on our homepage.
- All Materials and Methods need to be described in the main text. We would encourage you to use 'Structured Methods', our new Materials and Methods format. According to this format, the Materials and Methods section should include a Reagents and Tools Table (listing key reagents, experimental models, software and relevant equipment and including their sources and relevant identifiers) followed by a Methods and Protocols section in which we encourage the authors to describe their methods using a step-by-step protocol format with bullet points, to facilitate the adoption of the methodologies across labs. More

information on how to adhere to this format as well as downloadable templates (.doc or .xls) for the Reagents and Tools Table can be found in our author guidelines: . An example of a Method paper with Structured Methods can be found here:

- Please include a "Disclosure and Competing Interests Statement" in the main text.
- Please include a Data availability section describing how the data, code etc. have been made available. This section needs to be formatted according to the example below:
The datasets and computer code produced in this study are available in the following databases:
 - Chip-Seq data: Gene Expression Omnibus GSE46748 (<https://www.ncbi.nlm.nih.gov/geo/query/acc.cgi?acc=GSE46748>)
 - Modeling computer scripts: GitHub (<https://github.com/SysBioChalmers/GECKO/releases/tag/v1.0>)
 - [data type]: [full name of the resource] [accession number/identifier] ([doi or URL or identifiers.org/DATABASE:ACCESSION])
- For data quantification: please specify the name of the statistical test used to generate error bars and P values, the number (n) of independent experiments (specify technical or biological replicates) underlying each data point and the test used to calculate p-values in each figure legend. The figure legends should contain a basic description of n, P and the test applied. Graphs must include a description of the bars and the error bars (s.d., s.e.m.).
- When you resubmit your manuscript, please download our CHECKLIST (<https://bit.ly/EMBOPressAuthorChecklist>) and include the completed form in your submission.
Please note that the Author Checklist will be published alongside the paper as part of the transparent process (<https://www.embopress.org/page/journal/17444292/authorguide#transparentprocess>).

If you feel you can satisfactorily deal with these points and those listed by the referees, you may wish to submit a revised version of your manuscript. Please attach a covering letter giving details of the way in which you have handled each of the points raised by the referees. A revised manuscript will be once again subject to review and you probably understand that we can give you no guarantee at this stage that the eventual outcome will be favorable.

Kind regards,

Maria

Maria Polychronidou, PhD
Senior Editor
Molecular Systems Biology

We realize that it is difficult to revise to a specific deadline. In the interest of protecting the conceptual advance provided by the work, we recommend a revision within 3 months (9th Jan 2024). Please discuss the revision progress ahead of this time with the editor if you require more time to complete the revisions. Use the link below to submit your revision:

IMPORTANT: When you send your revision, we will require the following items:

1. the manuscript text in LaTeX, RTF or MS Word format
2. a letter with a detailed description of the changes made in response to the referees. Please specify clearly the exact places in the text (pages and paragraphs) where each change has been made in response to each specific comment given
3. three to four 'bullet points' highlighting the main findings of your study
4. a short 'blurb' text summarizing in two sentences the study (max. 250 characters)
5. a 'thumbnail image' (550px width and max 400px height, Illustrator, PowerPoint or jpeg format), which can be used as 'visual title' for the synopsis section of your paper.
6. Please include an author contributions statement after the Acknowledgements section (see <https://www.embopress.org/page/journal/17444292/authorguide>)
7. Please complete the CHECKLIST available at (<https://bit.ly/EMBOPressAuthorChecklist>).
Please note that the Author Checklist will be published alongside the paper as part of the transparent process (<https://www.embopress.org/page/journal/17444292/authorguide#transparentprocess>).
8. When assembling figures, please refer to our figure preparation guideline in order to ensure proper formatting and readability in print as well as on screen:

See also figure legend guidelines: <https://www.embopress.org/page/journal/17444292/authorguide#figureformat>

9. Please note that corresponding authors are required to supply an ORCID ID for their name upon submission of a revised manuscript (EMBO Press signed a joint statement to encourage ORCID adoption).
(<https://www.embopress.org/page/journal/17444292/authorguide#editorialprocess>)

Currently, our records indicate that the ORCID for your account is 0000-0002-6606-6884.

Please click the link below to modify this ORCID:
Link Not Available

The system will prompt you to fill in your funding and payment information. This will allow Wiley to send you a quote for the article processing charge (APC) in case of acceptance. This quote takes into account any reduction or fee waivers that you may be eligible for. Authors do not need to pay any fees before their manuscript is accepted and transferred to the publisher.

EMBO Press participates in many Publish and Read agreements that allow authors to publish Open Access with reduced/no publication charges. Check your eligibility: <https://authorservices.wiley.com/author-resources/Journal-Authors/open-access/affiliation-policies-payments/index.html>

*** PLEASE NOTE *** As part of the EMBO Press transparent editorial process initiative (see our Editorial at <https://dx.doi.org/10.1038/msb.2010.72>), Molecular Systems Biology publishes online a Review Process File with each accepted manuscripts. This file will be published in conjunction with your paper and will include the anonymous referee reports, your point-by-point response and all pertinent correspondence relating to the manuscript. If you do NOT want this File to be published, please inform the editorial office at msb@embo.org within 14 days upon receipt of the present letter.

Reviewer #1:

This manuscript reports on a large diallele eQTL experiment in Yeast in conjunction with long-read sequencing to assess the relative impacts of structural vs SNP variation on the expression variation. The inclusion of long-read sequencing is key to this work. While the results are intriguing, there is an issue of missed genomic variation (cytoplasmic) as well as issues surrounding the mapping of eQTL for accessory genes that need to be clarified.

I understand that the crossing design was non-reciprocal which limits the ability to investigate any cyto-nuclear interactions. However, is there no information on the mitochondrial genetic diversity in this panel that would allow an investigation of how the cytoplasmic genomic variation may be influencing the results. There are only at most 26 cytoplasmic haplotype and it would be possible to test how these 26 haplotypes are influencing the nuclear expression in manner similar to the GWAS analysis. To fully understand what the nuclear variation means on the expression on these genes, it seems necessary to provide some indication of how the cytoplasmic variation may be shaping the same expression patterns. For example, what is simply the ratio of nuclear to cytoplasmic heritability on expression patterns?

In Figure 2C, there are some distant large effect eQTL. In plants, it is becoming clear that these are actually genes that have moved laterally in the genome such that they are no longer in the reference position. And this ends up leading to a mis-identification of distant because the gene is now in the "distant" position. With the long-read sequencing was there any evidence of lateral gene movement that could explain these large effect distant eQTL? And it could lead to the report of accessory genes having larger effect distant eQTL.

I'm somewhat confused by the GWAS of the accessory genes. For 422 of these 708 accessory genes it was stated that they were variably present but it isn't clear what accessory means for the other 286 genes. From my understanding for accessory genes, there should be no expression in some of the genotypes as they are not present. Yet a majority of these do not find an eQTL? Is this because if a gene had no expression in a genotype, that the given genotype was discarded from the GWA (both SNP and SV)? From my understanding of the pangenome terminology, calling a gene an accessory indicates that it has presence/absence variation. This would indicate that in genotypes with absence there should be zero expression and lead to a large bimodal distribution for this genes transcript abundance that should identify a large local (possibly distant) eQTL. Yet this does not seem to have occurred from the reported analysis. This suggests a false-negative error rate where eQTL are being missed for these genes. One option is that the expression data was normalized to remove the zero expression genotypes. In other fungi like *Botrytis cinerea* this issue was due to unmapped SVs but the long-read genome sequence should eliminate this possibility. Some explanation for this lack of local cis-eQTL for genes that by default should have a local large-effect cis-eQTL should be provided.

In the SV analysis, variants present in only one of the 26 parents was excluded due to a concern about false positive signals. Was a similar methodology used for the SNPs? I could not quite tell from the methods or the results descriptions. If the goal is to compare SVs to SNPs, then it would seem that the same filtering approach should be used. Especially as the false detection issue should be similar.

Reviewer #2:

Tsouris et al investigated whether rare SNP and rare SV contribute to transcriptomic variation in populations. They did so by analyzing all-by-all hybrids from 26 parental isolates which were sequenced by several technologies, including long-read sequencing. They exploit the fact that shared ancestry is naturally broken up by recombination over evolutionary time to then perform variance partitioning and more importantly, GWAS for eQTL detection. They find that distant eQTLs have lower effect size than local eQTLs, that rare eQTLs have lower effect size than common variants, despite being more likely detected by their eQTL-detection approach and finally that SVs have lower effect size than SNPs.

Overall the paper investigates important questions that has been posed for decades, and resolves some of these. There are some expected results, but some which I was very surprised about. In my opinion, the fact that low-frequency variants have much lower effect on transcription should be discussed more heavily in light of the fact that low-frequency variants typically have more effect on growth. The authors do recognize this but make no further effort to explore this. This is very unexpected to me (see point 5).

I have a few thoughts:

- 1) The authors filter singletons from the SNP and SV data. This makes sense for various reasons, as all singletons would be linked together. But throughout the reading I was wondering whether the authors can estimate how much heritability the singletons contribute to. It seems perhaps the authors could do so by considering the singletons as a single locus and partitioning the variance accordingly.
- 2) I'm unable to find what the authors claim are SVs to the detail that I think is required. Is there a minimal size for the SV? Would an insertion of 1 nucleotide be an SV? I think this is important to specify since it allows us to interpret better why SVs have low effect size.
- 3) The authors should segregate the SVs from Ty-related elements with others and show the effect size of these two categories separately. I don't expect a Ty element to have much effect on anything unless it hops inside a gene or a promoter (and I believe the transposase usually avoids these regions), but I would expect a gene deletion to have massive effect on that gene's expression. This isn't something that seems to be captured here.
This is somewhat interesting because SVs have been shown to be very important in altering phenotypes of yeast strains (from pure missing pathways like sugar assimilation, or cation removal by *Ena1*). In those cases, the effect size was massive, so I'm curious how the authors can reconcile this (maybe an artifact of growing in rich media?).
- 4) Is there a distant/local relevant analysis with SVs that is possible? I wasn't able to find it here, but again it would be the first thing I would verify if some gene duplicated.
- 5) The fact that rare variants are enriched in 'causative phenotypic changes' is not surprising given the previous QTL papers on rare variants. However, what is surprising to me is the fact that their effect size is generally very low. This goes against the finding that these should have larger effect on growth and it's also not what we would expect purely from a statistical perspective since detectability is usually related to the effect (and to some extent to frequency of the variant in the pool). This led me to ponder what exactly was being plotted in 3D. Are the same variants plotted several times? I can see that there are only 104 low-frequency eQTL so I couldn't understand the $n = 1205$ as that is the number of low-frequency variants in the whole pool. Are rare-variants a random sampling of all the SNPs? Are they enriched in non-synonymous, synonymous, non-coding, specific gene function etc? I'm sure this is discussed in cited papers but a small mention of them here might re-contextualize some of the results discussed here.
- 6) The data description of '2,002,550 transcriptomic measurements' is very vague as it can relate to the number of transcriptomes rather than the number of genes with at least 1 read in all samples. I would recommend rephrasing.
- 7) There is a typo in Figure 3 (Frequeny instead of Frequency).
- 8) The color scheme for Figure 4a/b can be improved. The use of gradients for distinct categories is unusual and unfortunately it is impossible for me to tell which green is which part of the pie.

Reviewer #3:

Tsouris et al sequences the genomes of a panel of yeast hybrids and compare that to previously published transcriptome data to explore the genetic basis of variations in mRNA abundance. While the design is classical, the approach employed by the users give allow them to address important questions with fewer confounding effects or better power than some other studies. In particular, the experimental design used here allows the authors to test the contribution of both rare(r) SNPs and structural variants to variation in mRNA abundance. Doing so, they help explore some of the causes of the missing heritability, a well-known problem in the quantitative genetics field.

The main finding of the authors, that both rarer SNPs and structural variants contribute to variation in mRNA abundance but with

different magnitudes, is not entirely novel. Indeed, the significant contribution of rarer SNPs is quite well-established and has been the leading theme of several publications, including in yeast. The contribution of structural variation to traits is perhaps less well quantified in the published literature, but the fact that the authors detect relatively few associations of this type means that the generality and robustness of conclusions are somewhat questionable also in the current submission. The measured mRNA abundances correspond to one snapshot in time (optical density ~ 0.3 for a single environment, and it is quite possible that other time points or other environments would have resulted in different conclusions. There are very few efforts by the authors to extract a deeper understanding of the biology that underlies the observed patterns, and the study does not go beyond the quantification of statistical associations. For example, we do not learn much about what properties of low frequency variants that make them disproportionately likely to contribute to mRNA abundances. The authors do not trace down candidate causative SNPs nor do they validate some of these, or the contribution of candidate structural variants, which would typically be done in influential yeast GWAs papers. This could have revealed interesting biology and would have inspired confidence in that detected associations are real. In fact, as the mRNA abundance data was taken from a recent publication by the authors, the main new work in this paper is the long-read sequencing, and the GWAs. While competently performed here, long-read sequencing of many yeast genomes has been done since Yue 2017 and is now quite standard. The technical paragraph on the characteristics of structural variants in yeast therefore adds little new, and is not put into the perspective of what has been reported in previous studies. It would thus seem better suited to be in an extended Methods section. The Discussion has a helpful section on study limitations, but otherwise reiterates results and adds few perspectives or thoughts on why or how yeast, and other organisms, have evolved to produce the observed patterns of the genotype-phenotype maps. Overall, I see the submission as a mostly incremental advancement of the quantitative genetics field. I find it hard to justify its publication in MSB.

Minor comments: The authors harvest cells at around OD 0.3, which is interpreted as cells being in log-phase growth. But this really depends on the starting OD. Moreover, given the microcultivation format used, it's not evident, in absence of data, that a log-phase will occur at all. In this cultivation scale cell populations often start experiencing nutrient restrictions before all cells have exited the lag-phase. Probably, many cell populations are at or close to their peak growth rate at OD = 0.3, but the physiological state of cell populations at the time of harvest will probably differ a bit. This differences in physiological state may underlie some part of the associations observed, e.g. in terms of deviations of accessory genes. This is not to say that the associations are irrelevant, but as these difference in physiological state are likely to be temporary, it does cast some doubt on the extent to which conclusions can be generalized - across the life cycle of yeast, across environments and across species. The authors do exclude singletons from their analysis as these have a too-pronounced confounding population structure. But it wasn't clear to me to what extent they also account for the population structure of other rare variants with similar presence in a few strains.

Additional comments from referee cross-commenting

Reviewer #1

I would agree that the significance is somewhat hard to derive from the way that the manuscript is written. This is partly because it is almost exclusively about yeast with little discussion or introductory material laying out the broader picture across a wider array of organisms. This makes it a bit of effort for a generalist/non-yeast reader to have to work to place it in a broader context. This could be at least partly fixed by reworking to focus on a general quantitative genetics audience.

I think the conundrum of large effect rare variants controlling low amounts of variation is simply an issue of estimating r^2 in a population. The rare variant aspect will simply relegate the total variance to a lower realm.

I am less concerned about not tracking down individual causative genes and talking about them. At some level that would decrease the general biologists' interest as the specific gene would be unlikely to be broadly interesting across organisms. The phenomenology is where the general interest would lay.

I agree that there is some significant confusion about the SVs to expression and effect size.

Reviewer #2

Similar to Reviewer #1, I do not find a need for tracking down the causative genes or understanding the specifics of yeast biology for these quantitative genetics questions and I also do not think it is a fatal flaw to not be able to generalize to the whole cell cycle of an organism.

As for the conundrum that I have, I see the point that r^2 for rare variants is low if it's variance explained of the total population variation. However, I did not interpret the author's analysis to be r^2 . The authors specifically talk about effect size, which I interpret to be the actual difference in expression level. I'm pasting the relevant lines in their manuscript: ("low-frequency variants have a lower effect size compared to the common variants"), which directly contradicts another statement ("suggesting that low-frequency variants have a large impact on transcript abundance variation"). Perhaps this is something the authors can simply address by making their interpretation clearer. I do not like the analysis as presented. The fact that rare variants have lower population r^2 is not really surprising.

I do agree with Reviewer #3 that the main novelty stems from the long-read sequence, but the SV analysis could be improved as

I mentioned in the original review.

Finally, I would also agree that the significance of the work is hard to derive as written. However, I do believe that the broader quantitative genetics questions are important. Whether the advance is 'sufficient' is not really something I can comment on. The insight is perhaps limited, but I do not believe that the experiments have been performed before. While I could have guessed some of these results, I could not know before I read this and what I did not know was not yeast-specific.

Reviewer #3

I agree with reviewers 1 and 2 on the importance of the research question - but I remain cautious about the novelty of the submission. Four years ago, in eLife, both Bloom et al, and Fournier et al reported a disproportionate contribution of low-frequency variants to yeast fitness traits across a range of environments. This study uses a different experimental design and looks at mRNA abundances but does so in a single environment. It reports essentially the same conclusion.

There are also several publications reporting a, perhaps surprisingly, limited contribution of SVs to trait variation. Their lack of comprehensiveness when cataloguing SVs leaves some room for false negatives, and this, to me, is the most important motivation for the current study. But, I am not yet convinced that the authors make a major advance here, as their final, boiled-down list of SVs seems to be quite far from comprehensive and their conclusions are ultimately based on a handful of associations. The SV suggestions of reviewer 2 are prudent and may go some way to address this.

Beyond confirming previous reports that low-frequency variants contribute much and SVs much less to variation in mRNA abundance, I am not sure that the authors tell us much new. I would have expected some efforts on the side of the authors to shed further light on their primary observations. Are e.g. low-frequency variants particular in any way - in terms of what genetic features/functions they affect, how they affect them or where in the genome the affected features are situated. Asking for confirmation of causality is perhaps taking it too far, but at the very least I would expect a revised paper to do further statistical analysis along these lines - and to tell us a bit more about how the authors think around the why's, when's and how's of what seems to be important properties of the genotype-phenotype map.

Reviewer #1:

This manuscript reports on a large diallele eQTL experiment in Yeast in conjunction with long-read sequencing to assess the relative impacts of structural vs SNP variation on the expression variation. The inclusion of long-read sequencing is key to this work. While the results are intriguing, there is an issue of missed genomic variation (cytoplasmic) as well as issues surrounding the mapping of eQTL for accessory genes that need to be clarified.

I understand that the crossing design was non-reciprocal which limits the ability to investigate any cyto-nuclear interactions. However, is there no information on the mitochondrial genetic diversity in this panel that would allow an investigation of how the cytoplasmic genomic variation may be influencing the results. There are only at most 26 cytoplasmic haplotype and it would be possible to test how these 26 haplotypes are influencing the nuclear expression in manner similar to the GWAS analysis. To fully understand what the nuclear variation means on the expression on these genes, it seems necessary to provide some indication of how the cytoplasmic variation may be shaping the same expression patterns. For example, what is simply the ratio of nuclear to cytoplasmic heritability on expression patterns?

[R] The reviewer raised an important question about how cytoplasmic genome variation could impact gene expression. Unfortunately, our experimental design does not allow us to answer this for several reasons. First, unlike the nuclear genome, the mito genome in a hybrid does not correspond to the combined genotypes of the parents due to limited heteroplasmy and mitochondrial recombination in yeast (PMC4196626). As a result, the mito genotype cannot be easily inferred for our hybrid panel. Even using the RNAseq reads, the potential recombination breaks cannot be reliably detected as all the intergenic regions are not transcribed. Second, our diallele panel is consistent with batch crossed hybrids, where each individual is likely to have a different mitotype. For these reasons, we are unable to infer the mitochondrial genotypes for the hybrids and therefore cannot reliably estimate the contribution of mitochondrial genome variation to gene expression variation.

In Figure 2C, there are some distant large effect eQTL. In plants, it is becoming clear that these are actually genes that have moved laterally in the genome such that they are no longer in the reference position. And this ends up leading to a mis-identification of distant because the gene is now in the "distant" position. With the long-read sequencing was there any evidence of lateral gene movement that could explain these large effect distant eQTL? And it could lead to the report of accessory genes having larger effect distant eQTL.

[R] To check for this kind of potential lateral gene movements, we checked all *trans* eQTL hotspots (Figure 2B) genes and their localization in the long-read sequencing assemblies. All ORFs that overlapped with any eQTL hotspots (>10 associations for a given SNP) are located at the expected chromosomes and positions in the assemblies. We also checked the SNPs associated with top *trans*-eQTL with large effects (Figure 2C). In this case, because it's hard to localize single SNPs in the long-read assemblies, we examined whether there is any SVs that overlapped with these SNPs. We did not find any SVs at the same position as these associated SNPs. Therefore, the distant eQTL we observed here are unlikely to be due to lateral gene movements.

I'm somewhat confused by the GWAS of the accessory genes. For 422 of these 708 accessory genes it was stated that they were variably present but it isn't clear what accessory means for the other 286 genes.

[R] We are sorry about the confusion. The 708 accessory genes are the total number of accessory genes that are present in this panel according to previous annotations across the 1,011 natural isolates (PMC6784862). However, 286 out of these 708 genes are all present in parental strains and the hybrid panel. This concerns mostly the "Ancestral" category where the genes are absent only in relatively a small number of isolates in the 1,011 population. We redefined and clarified this in the revised manuscript. We are only focusing on the 422 genes as accessory in this hybrid panel. Changes are found in page 6, line 23-33.

From my understanding for accessory genes, there should be no expression in some of the genotypes as they are not present. Yet a majority of these do not find an eQTL? Is this because if a gene had no expression in a genotype, that the given genotype was discarded from the GWA (both SNP and SV)? From my understanding of the pangenome terminology, calling a gene an accessory indicates that it has presence/absence variation. This would indicate that in genotypes with absence there should be zero expression and lead to a large bimodal distribution for this genes transcript abundance that should identify a large local (possibly distant) eQTL. Yet this does not seem to have occurred from the reported analysis. This suggests a false-negative error rate where eQTL are being missed for these genes. One option is that the expression data was normalized to remove the zero expression genotypes. In other fungi like *Botrytis cinerea* this issue was due to unmapped SVs but the long-read genome sequence should

eliminate this possibility. Some explanation for this lack of local cis-eQTL for genes that by default should have a local large-effect cis-eQTL should be provided.

[R] For the 422 accessory genes that are relevant to this hybrid panel, the majority (307/422) corresponds to introgressions from a closely related *Saccharomyces* species, namely *S. paradoxus*. In these cases, both copies of the gene are present, albeit in a heterozygous configuration where one copy is from *S. cerevisiae* and the other copy is from *S. paradoxus*. Based on our other analysis on the same panel (<https://doi.org/10.1016/j.xgen.2023.100459>) and transcriptomics data across the 1,011 natural population (<https://doi.org/10.1101/2023.05.17.541122>), most of the introgressed genes have integrated the *S. cerevisiae* regulatory circuits and are not particularly enriched for cis regulation variation.

The remaining accessory genes mostly correspond to the “Unknown” category, where the origin of the ORF and/or their locations are not clear. We only have a small number of genes where we are certain that they are indeed presence/absence variation. These correspond to the horizontal gene transfer (HGT, 8 genes) genes that are only present in certain parent, therefore either completely absent in some hybrids or present as single copy. For this category, eQTL were mapped for 5 out of 8 HGT genes.

Overall, for the accessory genes, we found eQTL for 121/307 “Introgression” genes, 29/99 “Unknown” genes and 5/8 HGT genes. Globally, we tend to find proportionally more genes with any eQTL in the accessory genes than the core genes (1552/5770). Genes with presence/absence variation have the highest fraction with any detected eQTL. The effect sizes for genes with presence/absence variation are also significantly higher than other categories:

For the lack of local cis-eQTL, it is firstly due to an overall low number of genes with presence/absence variation across the dataset. Secondly, genes in the HGT and Unknown categories were originally annotated based on short reads sequencing data and therefore do not have a chromosomal location assigned. These eQTLs are therefore not classified as either local or distant.

We rewrote the text for the accessory gene analyses in the revised ms to clarify these points. Changes are found in page 6, line 19-21.

In the SV analysis, variants present in only one of the 26 parents was excluded due to a concern about false positive signals. Was a similar methodology used for the SNPs? I could not quite tell from the methods or the results descriptions. If the goal is to compare SVs to SNPs, then it would seem that the same filtering approach should be used. Especially as the false detection issue should be similar.

[R] Yes, the same filtering was applied for both SNPs and SVs. Descriptions for the filtering can be found in the method section.

Reviewer #2:

Tsouris et al investigated whether rare SNP and rare SV contribute to transcriptomic variation in populations. They did so by analyzing all-by-all hybrids from 26 parental isolates which were sequenced by several technologies, including long-read sequencing. They exploit the fact that shared ancestry is naturally broken up by recombination over evolutionary time to then perform variance partitioning and more importantly, GWAS for eQTL detection. They find that distant eQTLs have lower effect size than local eQTLs, that rare eQTLs have lower effect size than common variants, despite being more likely detected by their eQTL-detection approach and finally that SVs have lower effect size than SNPs.

Overall the paper investigates important questions that has been posed for decades, and resolves some of these. There are some expected results, but some which I was very surprised about. In my opinion, the fact that low-frequency variants have much lower effect on transcription should be discussed more heavily in light of the fact that low-frequency variants typically have more effect on growth. The authors do recognize this but make no further effort to explore this. This is very unexpected to me (see point 5).

I have a few thoughts:

1) The authors filter singletons from the SNP and SV data. This makes sense for various reasons, as all singletons would be linked together. But throughout the reading I was wondering whether the authors can estimate how much heritability the singletons contribute to. It seems perhaps the authors could do so by considering the singletons as a single locus and partitioning the variance accordingly.

[R] We rerun the hgln model dividing the variant matrix into singletons, all SNP without singletons and SV without singletons. The results are presented here:

On average, SNP without singletons accounts for 0.202 of the genome-wide heritability, compared to 0.062 for all singletons and 0.021 for SV without singletons.

2) I'm unable to find what the authors claim are SVs to the detail that I think is required. Is there a minimal size for the SV? Would an insertion of 1 nucleotide be an SV? I think this is important to specify since it allows us to interpret better why SVs have low effect size.

[R] SV are defined as variants that impact at least 50 bp. We added more descriptions in the revised text (Page 8, line 23-24).

3) The authors should segregate the SVs from Ty-related elements with others and show the effect size of these two categories separately. I don't expect a Ty element to have much effect on anything unless it hops inside a gene or a promoter (and I believe the transposase usually avoids these regions), but I would expect a gene deletion to have massive effect on that gene's expression. This isn't something that seems to be captured here.

[R] We segregated the SV-eQTL into according to their types, i.e. deletion (DEL), insertion (INS), contraction (CONTR) and duplication (DUP) and compared their effect sizes according to Ty or non-Ty related variants. The results are presented here:

Despite the overall low number of SV-eQTL, we do observe that deletions tend to have larger effect sizes than other types of variants. Moreover, Ty related variants tend to have smaller effect sizes than non-Ty related variants for deletions and insertions. The same comparison cannot be performed for contraction and duplication due to the low number of associated variants in these categories.

We included these results in the revised ms (page 9, line 28-32 and Figure S4D).

This is somewhat interesting because SVs have been shown to be very important in altering phenotypes of yeast strains (from pure missing pathways like sugar assimilation, or cation removal by *Ena1*). In those cases, the effect size was massive, so I'm curious how the authors can reconcile this (maybe an artifact of growing in rich media?).

[R] As the reviewer pointed out, there are many known examples of SVs with large effect size in yeast, including *MAL* genes for maltose assimilation, *SUL1* for sulfate transport, *CUP1* more copper resistance, and the *ENA* genes for sodium stress. However, all these cases are specific to a type of stress or growth conditions, and those genes are simply not expressed in the rich media where our experiment is performed. We believe the use of rich media is indeed why we do not observe such large effect size SV-eQTL.

4) Is there a distant/local relevant analysis with SVs that is possible? I wasn't able to find it here, but again it would be the first thing I would verify if some gene duplicated.

[R] There is only one local eQTL among the 20 SV-eQTL detected, which is an insertion (Ty-related) and has the largest effect size (0.039). Overall, duplication events are relatively rare across all SV types and many of these do not overlap with an entire ORF. Moreover, due to the nature of the diallel hybrids, most of these variants are present in a heterozygous state so the effects could be further buffered. A copy number variant (CNV) type of analyses based on

short-reads sequencing for the hybrid panel might be more appropriate to see the effect of deletion or duplication of genes on gene expression.

5) The fact that rare variants are enriched in 'causative phenotypic changes' is not surprising given the previous QTL papers on rare variants. However, what is surprising to me is the fact that their effect size is generally very low. This goes against the finding that these should have larger effect on growth and it's also not what we would expect purely from a statistical perspective since detectability is usually related to the effect (and to some extent to frequency of the variant in the pool).

[R] The reviewer raised an important point. This is a complicated question and we think there are currently not enough data to directly draw a link between expression traits and growth traits. One of the hypotheses we have is that previous QTL analyses on growth traits are focused on a diverse set of growth conditions whereas the expression traits are measured in rich media with no stress. However, more data is needed, for example RNAseq in different stress conditions, to really explain this apparent difference in growth traits vs. expression traits.

This led me to ponder what exactly was being plotted in 3D. Are the same variants plotted several times? I can see that there are only 104 low-frequency eQTL so I couldn't understand the n = 1205 as that is the number of low-frequency variants in the whole pool.

[R] This was indeed a mistake in the plot, the eQTL are accidentally plotted multiple times. The corrected numbers are 2534 for eQTL associated with common variants and 504 associated with low frequency variants. The 104 low-frequency eQTL referred to the number of unique loci associated, but not the total number of eQTL. We clarified all these numbers and corrected the figure in the revised ms (page 7 line 13-16).

Are rare-variants a random sampling of all the SNPs? Are they enriched in non-synonymous, synonymous, non-coding, specific gene function etc? I'm sure this is discussed in cited papers but a small mention of them here might re-contextualize some of the results discussed here.

[R] We annotated the full SNP matrix using SnpEff (PMC3679285). The results are summarized in the following table:

Annotation	Type	In_group	Total	Fraction	Fisher's test
Intergenic	Common	9686	30681	0.316	n.s
	LowFreq	412	1228	0.336	
initiator_codon_variant	Common	1	30681	0.0000326	n.s
	LowFreq	0	1228	0	
intron_variant	Common	228	30681	0.00743	n.s
	LowFreq	13	1228	0.0106	
missense_variant	Common	5297	30681	0.173	n.s
	LowFreq	204	1228	0.166	
non_coding_transcript_exon_variant	Common	83	30681	0.00271	n.s
	LowFreq	1	1228	0.000814	
splice_region_variant&intron_variant	Common	6	30681	0.000196	n.s
	LowFreq	0	1228	0	
splice_region_variant&stop_retained_variant	Common	23	30681	0.000750	n.s
	LowFreq	2	1228	0.00163	
splice_region_variant&synonymous_variant	Common	1	30681	0.0000326	n.s
	LowFreq	0	1228	0	
start_lost	Common	13	30681	0.000424	n.s
	LowFreq	0	1228	0	
stop_gained	Common	37	30681	0.00121	n.s
	LowFreq	2	1228	0.00163	
stop_lost&splice_region_variant	Common	7	30681	0.000228	n.s
	LowFreq	0	1228	0	
synonymous_variant	Common	15299	30681	0.499	n.s
	LowFreq	594	1228	0.484	

Across the full matrix, there is no difference in terms of variant types between common and low frequency variants. In terms of associated SNPs, same trend is observed:

Annotation	Type	In_group	Total	Fraction	Fisher's test
Intergenic	Common	332	918	0.362	n.s
	LowFreq	40	104	0.385	
intron_variant	Common	8	918	0.00871	n.s
	LowFreq	1	104	0.00962	
missense_variant	Common	188	918	0.205	n.s
	LowFreq	20	104	0.192	

synonymous_variant	Common	383	918	0.417	n.s
	LowFreq	43	104	0.413	

Overall, low frequency variants do appear to be a random sampling all SNPs. There is no significant difference in terms of mutation types between common and low frequency variants that are associated with any eQTL. We added some discussions about this point in the revised ms (page 7 line 24-36; page 10 line 22-31).

6) The data description of '2,002,550 transcriptomic measurements' is very vague as it can relate to the number of transcriptomes rather than the number of genes with at least 1 read in all samples. I would recommend rephrasing.

[R] We rephrased this sentence in the revised ms.

7) There is a typo in Figure 3 (Frequeny instead of Frequency).

[R] We corrected this in the revised ms.

8) The color scheme for Figure 4a/b can be improved. The use of gradients for distinct categories is unusual and unfortunately it is impossible for me to tell which green is which part of the pie.

[R] We changed the color scheme for this figure in the revised ms.

Reviewer #3:

Tsouris et al sequences the genomes of a panel of yeast hybrids and compare that to previously published transcriptome data to explore the genetic basis of variations in mRNA abundance. While the design is classical, the approach employed by the users give allow them to address important questions with fewer confounding effects or better power than some other studies. In particular, the experimental design used here allows the authors to test the contribution of both rare(r) SNPs and structural variants to variation in mRNA abundance. Doing so, they help explore some of the causes of the missing heritability, a well-known problem in the quantitative genetics field.

The main finding of the authors, that both rarer SNPs and structural variants contribute to variation in mRNA abundance but with different magnitudes, is not entirely novel. Indeed, the significant contribution of rarer SNPs is quite well-established and has been the leading theme of several publications, including in yeast. The contribution of structural variation to traits is perhaps less well quantified in the published literature, but the fact that the authors detect relatively few associations of this type means that the generality and robustness of conclusions are somewhat questionable also in the current submission. The measured mRNA abundances correspond to one snapshot in time (optical density ~0.3 for a single environment, and it is quite possible that other time points or other environments would have resulted in different conclusions.

[R] We agree that changing the culture conditions would possibly lead to condition specific conclusions. But here we are focusing on the mid-log growth cultures to have the same point of comparison with our previous population level RNAseq study (Caudal et al. 2023). Further studies are needed to see the specific effect of culture conditions on expression variation.

There are very few efforts by the authors to extract a deeper understanding of the biology that underlies the observed patterns, and the study does not go beyond the quantification of statistical associations. For example, we do not learn much about what properties of low frequency variants that make them disproportionately likely to contribute to mRNA abundances.

[R] We annotated the full SNP matrix using SnpEff and analyses the distribution of variant types (non-synonymous, synonymous, intron variants, non-sense variants etc.) across the full matrix as well as associated SNPs. There are no significant differences between common and low frequency variants for any of those features, suggesting that low frequency variants are a random sampling of all SNPs (See more details in response to reviewer 2, point 5). These comparisons are now included in the supplementary tables 10 and 11. We agree with the reviewer that the biology behind these observations is very important, but we are unable to answer them yet with the current dataset. We added results and discussions for this point (page 7 line 24-36; page 10 line 22-31).

The authors do not trace down candidate causative SNPs nor do they validate some of these, or the contribution of candidate structural variants, which would typically be done in influential yeast GWAs papers. This could have revealed interesting biology and would have inspired confidence in that detected associations are real.

[R] It would indeed be interesting to trace down some of the causative candidates. But as also pointed out by the other reviewers, our goal here is to have a global view of the impact of different variant types on gene expression. Nevertheless, the next step could be to focus on some specific cases to further explore specific biological questions.

In fact, as the mRNA abundance data was taken from a recent publication by the authors, the main new work in this paper is the long-read sequencing, and the GWAs. While competently performed here, long-read sequencing of many yeast genomes has been done since Yue 2017 and is now quite standard. The technical paragraph on the characteristics of structural variants in yeast therefore adds little new, and is not put into the perspective of what has been reported in previous studies. It would thus seem better suited to be in an extended Methods section.

[R] We think this paragraph is usual for the reader to contextualize the number and types of SVs in our diallel specifically. It's not intended to give an overview of the SVs in a natural population.

The Discussion has a helpful section on study limitations, but otherwise reiterates results and adds few perspectives or thoughts on why or how yeast, and other organisms, have evolved to produce the observed patterns of the genotype-phenotype maps. Overall, I see the submission as a mostly incremental advancement of the quantitative genetics field. I find it hard to justify its publication in MSB.

Minor comments: The authors harvest cells at around OD 0.3, which is interpreted as cells being in log-phase growth. But this really depends on the starting OD. Moreover, given the microcultivation format used, it's not evident, in absence of data, that a log-phase will occur at all. In this cultivation scale cell populations often start experiencing nutrient restrictions before all cells have exited the lag-phase. Probably, many cell populations are at or close to their peak growth rate at OD = 0.3, but the physiological state of cell populations at the time of harvest will probably differ a bit. This differences in physiological state may underlie some part of the associations observed, e.g. in terms of deviations of accessory genes. This is not to say that the associations are irrelevant, but as these difference in physiological state are likely to be temporary, it does cast some doubt on the extent to which conclusions can be generalized - across the life cycle of yeast, across environments and across species.

[R] We used mid-log growth culture (OD ~0.3) that we previously established for high-throughput RNAseq (Caudal et al. 2023).

The authors do exclude singletons from their analysis as these have a too-pronounced confounding population structure. But it wasn't clear to me to what extent they also account for the population structure of other rare variants with similar presence in a few strains.

[R] For the variant matrix, we only removed the singletons. To account for population structure, we used LMM based GWAS with the genotype matrix as kinship, which should account for such effects.

24th Jan 2024

Manuscript Number: MSB-2023-12009R

Title: Diallel panel reveals a significant impact of low-frequency variants on gene expression variation

Dear Joseph,

Thank you for sending us your revised manuscript. We have now heard back from the three reviewers who were asked to evaluate your revised study. As you will see below, the reviewers are satisfied with the performed revisions and support publication. Reviewer #1 recommends adding some of the important discussion points (currently only in the point-by-point response) in the manuscript itself. We would ask you to perform these changes in a minor revision. We would also ask you to address some editorial issues listed below.

- Our Data Editors noted that the following needs to be corrected/added in the Figure Legends:

-- Please define the annotated p value *** in the legend of supplementary figure 2d; as appropriate.

-- Please indicate the statistical test used for data analysis in the legends of supplementary figures 1c-e; 3b.

-- The box plots need to be defined in terms of minima, maxima, centre, bounds of box and whiskers, and percentile in the legends of figures 2c; 3c; 4c; supplementary figure 2d.

-- Please include information related to n in the legends of figure 3c; supplementary figures 2d; 4d.

-- Please describe the nature of entity for 'n' (technical? biological?) in the legends of figures 2c; 4c.

-- Please define the error bar in the legend of supplementary figure 4d.

- The funding information provided in the manuscript text should match the information entered in the online submission system. The information "Interdisciplinary Thematic Institutes (ITI) Integrative Molecular and Cellular Biology (IMCBio), as part of the ITI 2021-to-2028 program of the University of Strasbourg, CNRS, and Inserm" is missing from the submission system.

- The keywords need to be reduced to 5.

- Please remove the 'Authors Contributions' from the manuscript. The 'Author Contributions' section is replaced by the CRediT contributor roles taxonomy to specify the contributions of each author in the journal submission system. Please use the free text box in the 'author information' section of the online submission system to provide more detailed descriptions if needed (e.g., 'X provided intracellular Ca⁺⁺ measurements in fig Y').

- Please include page numbers in the Appendix Table of Contents. Appendix Figures should be labelled and called out as Appendix Figure S1-S4 (NOT Supplementary Figures S1-S4).

- Tables S1-S11 should be provided and called out in the text as Datasets EV1-EV11. Please provide one file (.xls or .csv) per EV Dataset. Please include the description of each EV Dataset in the dataset file itself, i.e. in a separate tab for .xls files or as a README.txt file in .zip folders for .csv files. The descriptions of the Datasets and the section "online data files" should be removed from the Appendix.

- Please include callouts to all EV Datasets in the text.

- The Figure legends should be placed after the References.

- Please update the reference to Tsouris et al, 2024 to that of the published paper in Cell Genomics.

Please resubmit your revised manuscript online, with a covering letter listing amendments and responses to each point raised by the referees. Please resubmit the paper ****within one month**** and ideally as soon as possible. If we do not receive the revised manuscript within this time period, the file might be closed and any subsequent resubmission would be treated as a new manuscript. Please use the Manuscript Number (above) in all correspondence.

Click on the link below to submit your revised paper.

Kind regards,

Maria

Maria Polychronidou, PhD
Senior Editor
Molecular Systems Biology

If you do choose to resubmit, please click on the link below to submit the revision online before 23rd Feb 2024.

IMPORTANT:

Please note that corresponding authors are required to supply an ORCID ID for their name upon submission of a revised manuscript (EMBO Press signed a joint statement to encourage ORCID adoption).

(<https://www.embopress.org/page/journal/17444292/authorguide#editorialprocess>)

Currently, our records indicate that the ORCID for your account is 0000-0002-6606-6884.

Link Not Available

*** PLEASE NOTE *** As part of the EMBO Press transparent editorial process initiative (see our Editorial at <https://dx.doi.org/10.1038/msb.2010.72> , Molecular Systems Biology will publish online a Review Process File to accompany accepted manuscripts. When preparing your letter of response, please be aware that in the event of acceptance, your cover letter/point-by-point document will be included as part of this File, which will be available to the scientific community. More information about this initiative is available in our Instructions to Authors. If you have any questions about this initiative, please contact the editorial office (msb@embo.org).

Reviewer #1:

My previous questions have been addressed. However, it might be good to include some of those discussion points in the manuscript itself. This would include some admission that the missing mitochondrial genetic information is constraining the ability to parse heritability to the different genomes. Similarly, the fact that presence/absence variation in other systems like plants or pathogenic fungi (see *Nuerospora*, *Scleoritinia*, *Verticillium*, *Botrytis*, etc) is much higher than in Yeast or humans means that the fractions reported here may not be directly relevant to genome evolution in those systems. Presently, these caveats are not clearly conveyed to the reader.

Reviewer #2:

I believe the authors have adequately clarified the important points I previously raised.

Reviewer #3:

The authors have engaged with and responded well to most of my comments. I am still missing some analysis to convince me that detected eQTLs capture real biology (which should have been possible given e.g. the numerous known links between transcription regulation and the expression of specific genes in yeast) and that conclusions have a broader validity (outside of OD 0.3 yeast growth in rich medium). To me, this reduces the impact and significance of the paper a bit. But the authors have generally done a good job, considering the limitations of their datasets, and I see no major reason to delay the publication of this paper.

All editorial and formatting issues were resolved by the authors.

30th Jan 2024

Manuscript number: MSB-2023-12009RR

Title: Diallel panel reveals a significant impact of low-frequency variants on gene expression variation

Dear Joseph,

Thank you again for sending us your revised manuscript. We are now satisfied with the modifications made and I am pleased to inform you that your paper has been accepted for publication.

Kind regards,

Maria

Maria Polychronidou, PhD
Senior Editor
Molecular Systems Biology
